



# Reconstruction of Holocene oceanographic conditions in the
# Northeastern Baffin Bay
*Katrine Elnegaard Hansen[1], Jacques Giraudeau[2], Lukas Wacker[3], Christof Pearce[1], Marit-Solveig*
*Seidenkrantz[1]*
[1]Department of Geoscience, Arctic Research Centre and iClimate, Aarhus University
[2]Université de Bordeaux, CNRS
[3]ETH, Zürich
*Correspondence to:* Katrine Elnegaard Hansen (katrine.elnegaard@geo.au.dk)
**Abstract**
The Baffin Bay is a semi-enclosed basin connecting the Arctic Ocean and the western North
Atlantic, thus making out a significant pathway for heat exchange. Here we reconstruct the
alternating advection of relatively warmer and saline Atlantic waters versus the incursion of colder
Arctic water masses entering the Baffin Bay through the multiple gateways in the Canadian Arctic
Archipelago and the Nares Strait during the Holocene. We carried out benthic foraminiferal
assemblage analyses, X-Ray Fluorescence scanning and radiocarbon dating of a 738 cm long
marine sediment core retrieved from the eastern Baffin Bay near Upernavik (Core AMD14-204C;
987 m water depth). Results reveal that the eastern Baffin Bay was subjected to several
oceanographic changes during the last 9.2 ka BP. Waning deglacial conditions with enhanced
meltwater influxes and an extensive sea-ice cover prevailed in the eastern Baffin Bay from 9.2-7.9
ka BP. A transition towards bottom water ameliorations are recorded at 7.9 ka BP by increased
advection of Atlantic water masses, encompassing the Holocene Thermal Maximum. A cold
period with growing sea-ice cover at 6.7 ka BP interrupts the overall warm subsurface water
conditions, promoted by a weaker northward flow of Atlantic waters. The onset of the
Neoglaciation at ca. 2.9 ka BP, is marked by an abrupt transition towards a benthic fauna
dominated by agglutinated species likely partly explained by a reduction of the influx of Atlantic
water, allowing increased influx of the cold, corrosive Baffin Bay Deep Water originating from
the Arctic Ocean, to enter the Baffin Bay through the Nares Strait. These cold subsurface water
conditions persisted throughout the late Holocene, only interrupted by short-lived warmings
superimposed on this cooling trend.



## 1 Introduction

The opening of the Nares Strait and the narrower gateways of the Canadian Arctic Archipelago (CAA) was initiated towards the end of the last glacial. It was completed in the Early Holocene at 9.3-8.3 ka BP, when parts of the Greenland and Innuitian ice sheets, blocking these gateways, had fully retreated from the area (Jennings et al., 2019; Georgiadis et al., 2018; Jennings et al., 2011; England et al., 2006; Zreda et al., 1999). The opening of these gateways presumably had a significant impact on the general oceanic circulation in Baffin Bay and the Labrador Sea, allowing the input of cold Arctic water masses to these regions (Jennings et al., 2019; Jennings et al., 2017).

The modern marine environment of Baffin Bay is characterised by a combination of warm Atlantic and cold polar waters. The West Greenland Current (WGC), which flows northward along the coast of West Greenland, carries mixed warm Atlantic-sourced Irminger Current Water and cold and fresh waters of the East Greenland Current (Drinkwater, 1996). The WGC is therefore a major source of Atlantic waters to Baffin Bay, transporting warm and saline water masses to high latitudes. The onset of the present configuration of the WGC during the late glacial (Jennings et al., 2017; Jennings et al., 2018) enabled the advection of Atlantic-sourced waters from the south along the west coast of Greenland into Baffin Bay. These waters progressively expanded from the shelf edge to shallow shelf areas during the deglaciation following the retreat of the Greenland ice-sheet (Jennings et al., 2017; Sheldon et al., 2016). Today, Atlantic water reaches the locations of Thule (76°N) and the southern part of the Nares Strait at its northernmost extension off West Greenland (Buch, 1994; Funder, 1990; Knudsen et al., 2008).

Water masses originating from the Arctic Ocean flow southward in the western part of the Baffin Bay (Baffin Current, Fig. 1A), where they act as a substantial contributor of freshwater to the Labrador Sea (Aksenov et al., 2010; Bunker, 1976; Yang et al., 2016). This influence of both cold Polar and warm Atlantic water masses makes Baffin Bay an important area of water mass exchange. Fluctuations in the entrainment of these fresh Polar water masses into the Labrador Sea have been suggested to influence the deep-water formation in the Labrador Sea and thus the Atlantic Meridional Overturning Circulation (AMOC) (Jones and Anderson, 2008; Sicre et al., 2014); consequently, they act as a key element in global heat transport. An increased entrainment of Irminger Current water masses into the WGC leads to local increased air temperatures and contributes to the retreat of marine outlet glaciers of West Greenland facilitated by submarine and surface melting, causing local freshening (Andresen et al., 2011; Jennings et al., 2017).



Furthermore, ocean and atmospheric forced melting can contribute to a speed up of the marine
outlet glaciers and general instability of the ice dynamics (Holland et al, 2008; Rignot et al, 2010;
Straneo & Heimbach, 2013; Straneo et al., 2013).
Several studies suggest that the eastern Baffin Bay has been subjected to a series of oceanographic
and paleoclimatic changes during the Holocene, induced by changes in the strength of the WGC
linked to fluctuations in Atlantic water entrainment and thus to changes in the AMOC. Most of
these studies focused on the southern and central shelf regions of West Greenland (Erbs-Hansen
et al., 2013; Moros et al., 2015; Lloyd et al., 2007; Perner et al., 2013; Seidenkrantz et al., 2007),
but fewer investigated the past dynamics of the WGC  in the northeastern sector of Baffin Bay.
In this study, we investigate potential changes in the influx of Atlantic-sourced water to the eastern
Baffin Bay through the Holocene, discussing the hypothesis that changes in Baffin Bay
environmental conditions are closely linked to overall changes in the Atlantic Meridional
Overturning Circulation (AMOC). Our study is based on micropalaeontological and geochemical
investigations of a marine sediment core retrieved near Upernavik in the Eastern Baffin Bay. This
site is located in the flow path of the WGC and in the vicinity of the marine outlet glacier Upernavik
Isstrøm (Fig. 1B). Faunal assemblage analysis of benthic foraminifera, radiocarbon datings and X-
ray Fluorescence (XRF) data enable the reconstruction of the palaeoceanography and paleoclimate
of the northeastern Baffin Bay, including the temporal and spatial development of the water
exchange in Baffin Bay during the Holocene.

**1.1 Regional setting**

The Baffin Bay is a semi-enclosed basin constrained by the Baffin Island to the west, Ellesmere
Island to the northwest and Greenland to the east (Fig. 1A). The basin is linked to the Atlantic
Ocean via the Labrador Sea and the 640 m deep and 320 km wide Davis Strait sill in the south,
and is connected to the Arctic Ocean through shallow gateways: Lancaster Sound (125 m deep)
and Jones Sound (190 m deep) to the northwest and the deeper Nares Strait (250 m deep) to the
north (Tang et al., 2004) (Fig. 1A). The open connections between the Arctic Ocean and Labrador
Sea/North Atlantic Ocean makes the Baffin Bay an important area for Polar water export and water
mass exchange with the North Atlantic Ocean. The mean water depth in Baffin Bay is <800 m,
where the deepest point of the bay in the large central abyssal region exceeds 2300 m water depth
(Tang et al., 2004; Welford et al., 2018). An area of maximum 80,000 km$^2$ in the northwestern
Baffin Bay is occupied by the North Water Polynya (Dunbar & Dunbar, 1972; Tremblay et al.,





2002). The prevailing northwesterly winds carry newly formed sea ice away from the polynya,
limiting the formation of a thick sea-ice cover resulting in open water conditions, extensive heat
loss to the atmosphere and high marine productivity (Melling et al., 2010). The sea ice that is
exported from the polynya contributes to brine formation, which may lead to sinking of dense and
cold surface waters. The sustainment of the polynya is highly dependent on strong northwesterly
winds and the continuous formation of an ice bridge at Smith Sound (Fig. 1A) preventing sea ice
from entering Baffin Bay through Nares Strait (Dunbar & Dunbar, 1972; Melling et al., 2010).
The modern ocean surface circulation in Baffin Bay is driven by the local atmospheric circulation
system affecting the strength of the northwesterly winds, creating an overall cyclonic ocean
circulation pattern (Drinkwater, 1996) (Fig. 1A). From the south near Cape Farewell, the mixed
WGC carries relatively warm saline water from the Irminger Current (IC) and cold ice loaded
Polar waters from the East Greenland Current (EGC) towards the north over the shelf region of
the West Greenland margin (Drinkwater, 1996), creating the West Greenland Intermediate Water
(Tang et al., 2004). The IC water component is mainly constrained to the continental slope in the
depth range of 200-1000 m, whereas the EGC component is more shelf oriented and thus shallower
(200 m), (Buch, 1994; Rykova et al., 2015). The WGC bifurcates into two branches when reaching
Davis Strait (Cuny et al., 2002). Here, one branch flows towards the west and eventually meets
and joins the Outer Labrador Current and heads south (Cuny et al., 2002; Drinkwater, 1996). The
other WGC branch continues northward along the west coast of Greenland and at turns westwards
at 75 °N, where it mixes with Arctic waters entering the Baffin Bay from the north through Nares
Strait and the gateways in the Canadian Arctic Archipelago (CAA) (Drinkwater, 1996). These
combined water masses make up the Baffin Current (BC), which comprises a major part of the
freshwater content in the southward flowing Labrador Current (Mertz et al., 1993). Parts of the
surface outflow from the CAA gateways recirculate eastward to the northeastern Baffin Bay
(Landry et al, 2015). The relative contribution of water masses from the IC and EGC plays a
prominent role in the temperature and salinity signature of the WGC.



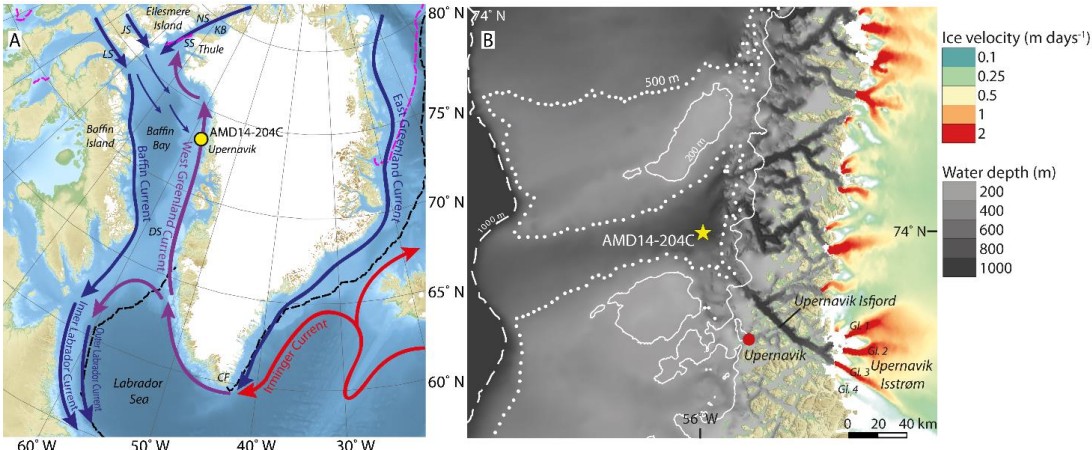


**Figure 1** A: Map showing the study site and the modern ocean surface circulation. Red, blue and purple arrows represents warmer, colder and mixed/intermediate water temperatures, respectively. The core AMD14-204C is marked with the yellow circle. The pink and black dashed lines mark the median sea-ice extent from 1981-2010 in September and March respectively (NSIDC, 2019). Abbreviations: LS = Lancaster Sound, JS = Jones Sound, NS = Nares Strait, SS = Smith Sound, KB = Kane Basin, DS= Davis Strait, CP = Cape Farewell. B: Close up on the Upernavik Isstrøm area, showing the local bathymetry and ice stream velocities. The Upernavik Isstrøm is comprised by four glaciers. The ocean bathymetry and bed topography data is derived from GEBCO (Weatherall et al., 2015) and BedMachine v3(Morlighem et al., 2017) and the ice stream velocity data is derived from Sentinel-1 SAR data acquired from 2017-12-28 to 2018-02-28 (Nagler et al., 2015). Abbreviations: Gl. = glacier.

The deeper part of the Baffin Bay (1200-1800 m water depth) is subjected to the cold, saline Baffin Bay Deep Water (BBDW). Water masses at depths exceeding 1800 m are referred to as Baffin Bay Bottom Water (BBBW) (Tang et al., 2004) (Fig. 2). Several hypotheses for the source of these water masses include local brine production in connection with winter sea ice formation on the shelf (Tan & Strain, 1980), cooled subsurface waters from Kane Basin flowing in via Nares Strait in a pulse like manner (e.g. Aksu, 1981; Collin, 1965), and the migration of cold, saline waters produced at the North Water Polynya (Bourke & Paquette, 1991).

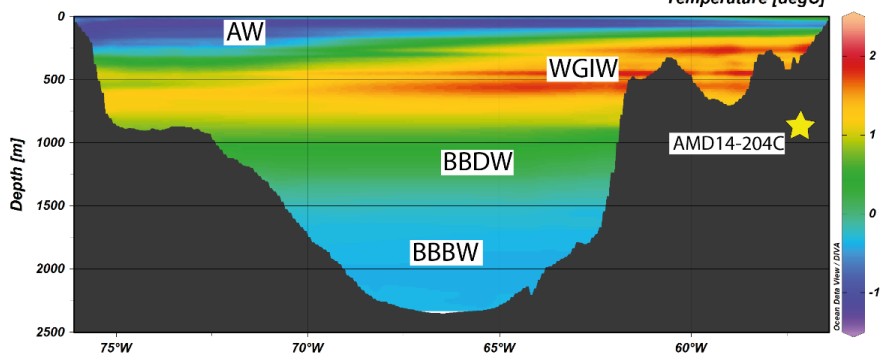


**Figure 2:** Water temperature transect at 73°N showing the different water masses. The yellow star indicate the core site at 987 m water depth. Abbreviations: AW = Arctic Water (0-300 m), WGIW = West Greenland Intermediate Water (300-800 m), BBDW =



Baffin Bay Deep Water (1200-1800 m), BBBW = Baffin Bay Bottom Water (1200-1800 m), (Tang et al., 2004). Temperature data
from World Ocean Atlas (Locarnini et al., 2013).
The modern sea-ice duration in Baffin Bay is longest in its north-western sector, and shortest in its
eastern region influenced by the northward flow of the warmer WGC (Tang et al., 2004; Wang et
al., 1994). Sea ice starts forming in open waters in the north and most of the bay is fully covered
by sea ice by March. In September, sea ice is limited to the CAA, and Baffin Bay is primarily
influenced by a sporadic thinner sea-ice cover (Tang et al., 2004) (Fig. 1A).
The shelf region of West Greenland is incised by numerous canyons and fjords among which
Upernavik Isfjord is the nearest to our core site (Fig. 1B). The fast-flowing marine-based outlet
glaciers that make up Upernavik Isstrøm terminate in the Upernavik Isfjord (Fig. 1B) (Briner et
al., 2013). Results from previous studies suggest that retreats of the ice stream are influenced by
the advection of warmer Atlantic waters into the fjord (Andresen et al., 2014; Vermassen et al.,

154   2019).

**2   Material and methods**
The presented multiproxy study is based on the analysis of marine sediment core AMD14-204C,
a Calypso Square (CASQ) gravity core collected on board the Arctic research vessel, Canadian
Coast Guard Ship (CCGS) *Amundsen* as part of the ArcticNet leg 1b expedition in 2014. The 738
cm long core was retrieved from 987 m water depth, in northeastern Baffin Bay (73°15.663'
N/57°53.987' W) at the head of the Upernavik Trough near Upernavik Isstrøm (Fig. 1). Shortly
after retrieval, the 738 cm long gravity core was subsampled into five core sections on board the
research vessel using 150 cm-long U-channels. These were subsequently kept in cold storage.

**2.1 Chronology**
The age model for core AMD14-204 C is based on 11 AMS (Accelerator Mass Spectrometry)
radiocarbon dates, mainly consisting of mixed benthic foraminiferal species. One sample also
contains some mixed ostracod species and two samples encompass both benthic and planktonic
foraminifera due to the scarcity of calcareous material in the core, see Table 1. Four of these mixed-
species radiocarbon dates have previously been used in an earlier version of the age model (Caron
et al., 2018;), and our revised age model includes seven additional levels of radiocarbon dates
measured at the ETH Laboratory, Ion Beam Physics in Zürich, see Table 1 and Supplementary for
further details on the method. These latter samples are based on either pure benthic or pure
planktonic species; for four of the levels we could date both samples based on benthic and on





planktonic specimens, where only the samples with benthic species were used in the age model.
All conventional radiocarbon ages were calibrated using the Marine13 radiocarbon calibration data
(Reimer et al., 2013) with the OxCal v4.3 software (Ramsey, 2008). A marine reservoir correction
of ΔR = 140±30 years has previously been used in similar studies of the Baffin Bay and west
Greenland area (e.g. Lloyd et al., 2011, Perner et al., 2012, Jackson et al., 2017) and is therefore
used in the calibration of the radiocarbon dates in this study.
**Table 1**: List of radiocarbon dates and modelled ages in core AMD14-204C. The dates with a * sign have previously been published
in Caron et al., 2018. All dates were calibrated using the Marine13 calibration curve (Reimer et al 2013) and ΔR = 140 ± 30 years.

| Sample depth midpoint (cm) | Lab. ID | Material | $^{14}C$ age (yr BP) | Calibrated age range (cal yr. BP), 1σ | Modelled median age (cal. yr BP) |
|---|---|---|---|---|---|
| 4.5 | ETH-92277 | Mixed benthic foraminifera | 705±50 | 167-276 | 213 |
| 70.5 | ETH-92279 | Mixed benthic foraminifera | 1795±50 | 1175-1270 | 1216 |
| 70.5 | ETH-92278 | Mixed planktonic foraminifera | 1710±50 | 1032-1175 | 1101 |
| 170* | SacA 46004 | Mixed benthic & planktonic foraminifera | 3555±35 | 3139-3260 | 3192 |
| 250.5* | BETA 467785 | Mixed benthic & planktonic foraminifera | 4300±30 | 4133-4254 | 4199 |
| 310.5 | ETH-92281 | Mixed benthic foraminifera | 4950±60 | 4860-4992 | 4941 |
| 310.5 | ETH-92280 | Mixed planktonic foraminifera | 4940±70 | 4930-5188 | 5043 |
| 410.5 | ETH-92283 | Mixed benthic foraminifera | 5805±60 | 5905-6005 | 5959 |
| 410.5 | ETH-92282 | Mixed planktonic foraminifera | 5825±60 | 5984-6155 | 6063 |
| 501.5* | BETA 488641 | Mixed benthic foraminifera | 6400±30 | 6656-6751 | 6707 |
| 580.5 | ETH-92285 | Mixed benthic foraminifera | 7155±70 | 7430-7531 | 7483 |
| 580.5 | ETH-92284 | Mixed planktonic foraminifera | 7005±60 | 7298-7417 | 7356 |
| 610* | SacA 46005 | Mixed benthic foraminifera & ostracods | 7445±50 | 7712-7822 | 7766 |
| 700.5 | ETH-92286 | Mixed benthic foraminifera | 8270±389 | 8639-8885 | 8755 |
| 737.5 | ETH-92287 | Mixed benthic foraminifera | 8489±154 | 9017-9302 | 9162 |


### 2.2 Foraminifera

Sediment samples of 1 cm width were subsampled every 10 cm throughout most of the core for
foraminiferal analyses, except for the 500-503 cm interval, the top (4-5cm) and bottom (737-738
cm) of the core where every 1 cm was counted and subsequently used for radiocarbon dating. The



wet sediment samples were weighed followed by wet sieving using sieves with mesh sizes of 0.063, 0.100 and 1 mm. Each fraction was dried in filter paper in the oven at 40 °C overnight before they were weighed and stored in glass vials. For the benthic foraminiferal assemblage analyses, the 0.063 and 0.100 mm fractions were combined, and both calcareous and agglutinated species were identified and counted together in order to reach sufficient total counts for reliable assemblage analyses. In all cases we were able to identify at least 300 benthic individuals, following the method used in (Lloyd et al., 2011; Perner et al., 2011; Perner et al., 2012).

**2.3 X-ray Fluorescence**

The non-destructive X-ray Fluorescence (XRF) method allows the measurement of changes in the bulk geochemical elemental compositions of the core without disturbing the sediment. The core was scanned and logged in 5 mm steps using an AVAATECH scanner at the EPOC laboratory in Bordeaux. The scan was conducted with generator settings of 10, 30 and 50 kV using a Rhodium (Rh) tube in order to get the full elemental spectra from Al to Ba. Data have previously been presented by Giraudeau et al., (submitted).

**3   Results**

**3.1 Core description**

The core primarily consists of hemipelagic mud. The lowermost part of the core (738-610 cm) is composed of greyish brown (2.5 Y/4/2) homogenous clayey silt, transitioning to bioturbated, olive grey (5Y 4/2) clayey silt in the upper part of the core (Caron et al., 2018).

**3.2 Chronology**

In previous studies of Core AMD14-204C (Caron et al., 2018; Giraudeau et al., submitted) age models were based on radiocarbon dating of bulk sediment samples, and paleomagnetic markers, with only a few foraminifera $^{14}$C dates. Our present study includes several new radiocarbon dates on foraminifera, and therefore no longer includes the bulk datings. Our 11 calibrated $^{14}$C dates, primarily based on foraminifera, reveal that the 738 cm-long sediment core encompasses the last ca. 9200 cal. years BP, covering most of the Holocene (Fig. 3). For the age depth modelling, a depositional P_sequence model was used with a k-value of 0.68 (Ramsey, 2008). The average sedimentation rate for the core is 86 cm/k year.

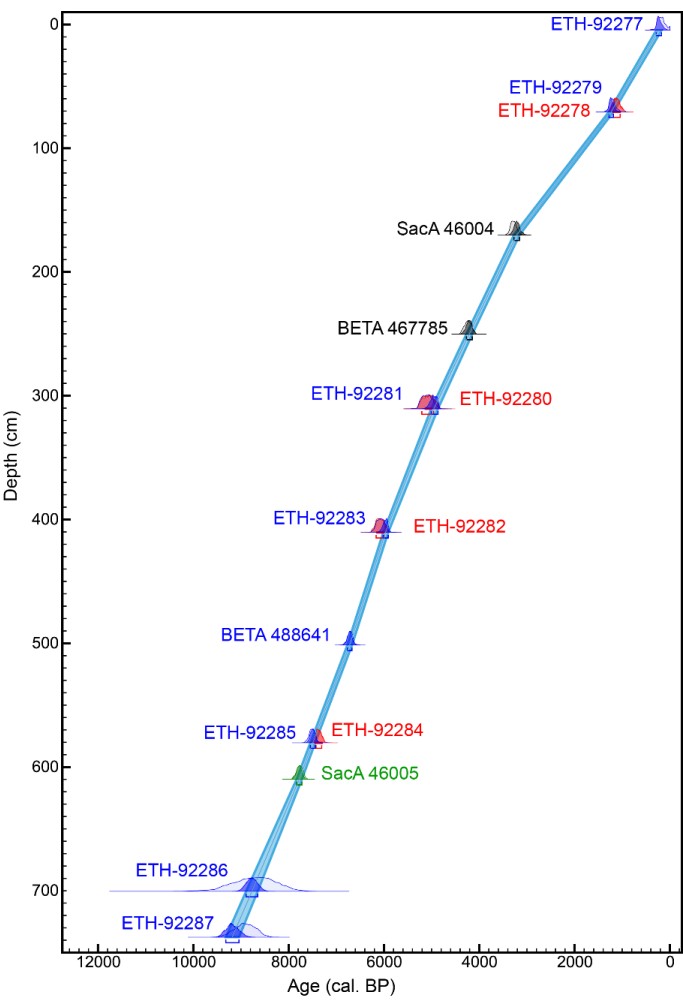

**Figure 3:** Age model for core AMD14-204C based on 11 radiocarbon dates (the green, black and blue dates). The light blue envelope represents the modelled 1σ range, and the blue line marks the modelled median age. The light shaded areas for each radiocarbon date indicate the probability distribution prior age modelling whereas the darker areas indicate the posterior probability distribution. Blue; mixed benthic foraminifera, red; mixed planktonic foraminifera, grey; mixed planktonic and benthic foraminifera, green; mixed ostracods, planktonic and benthic foraminifera.

Pairs of mixed benthic and mixed planktonic calibrated $^{14}C$ dates measured at the same sample depths 70.5, 310.5, 410.5 and 580.5 show only small differences (Fig. 3), all of which lie within the same age uncertainty. These results suggest that the radiocarbon ages measured from samples of mixed benthic and planktonic species are reliable. Today, the water carried by the WGC occupies the whole water column over the continental margin of eastern Baffin Bay (Cuny et al., 2002; Tang et al., 2004). The similar dates obtained from pairs of planktonic and benthic foraminifera specimens in samples from the top to the bottom part of the core suggest that, at our





study site, the subsurface and bottom waters were subjected to the same water mass throughout
the Holocene.

### 3.3 Foraminifera

The agglutinated and calcareous benthic foraminiferal tests were in general well preserved
throughout the core and there were minor signs or no signs of post mortem dissolution of the tests.
A total of 43 calcareous and 17 agglutinated benthic foraminiferal taxa were identified. The
relative abundances in percent were calculated from the entire benthic foraminiferal assemblages
(combined agglutinated and calcareous foraminiferal specimens assemblage to allow statistically
sufficient count numbers), and the benthic species shown in the figures all have a percentage
frequency of 4 % in at least one of the sample intervals of the core (Fig. 4 and 5). Planktonic
foraminiferal specimens are on average 10 times less abundant than benthic specimens, with the
lowest abundance at the bottom of the core. A down core succession of four ecozones was defined
based on major changes in the relative abundance of the most abundant benthic species, indicative
of changes in the environment.

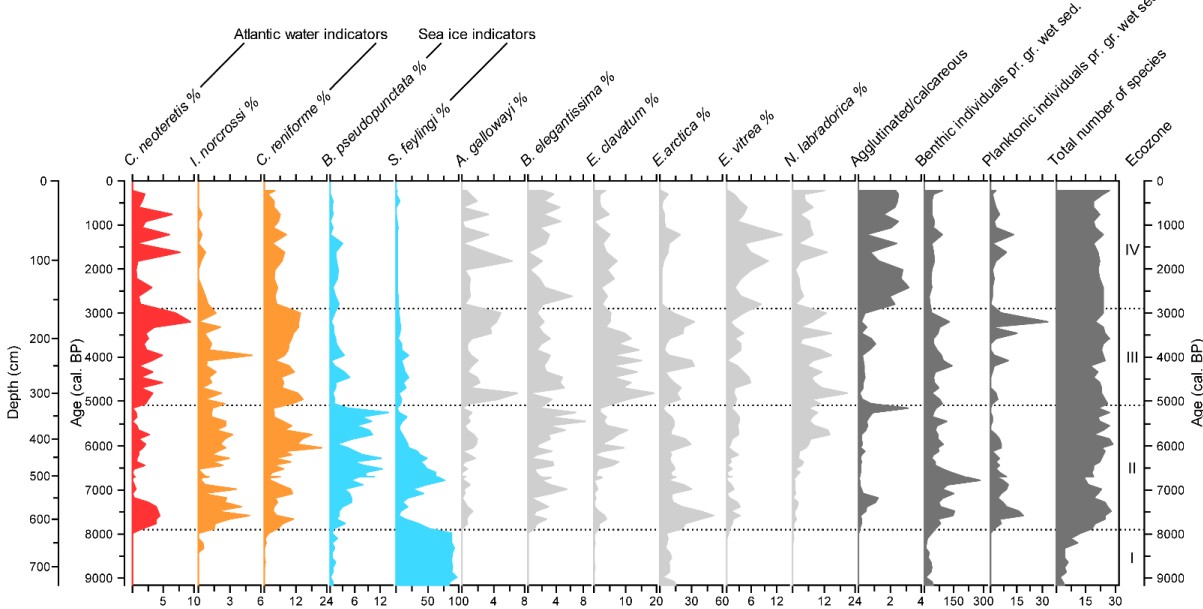

**Figure 4:** Downcore distribution of the most abundant (>4% in at least one sample) calcareous benthic foraminiferal species.
Ecozones (I to IV) are shown on the right side of the figure. Relative abundances are calculated based on the entire benthic
(calcareous and agglutinated) foraminiferal assemblage. Some species are grouped (colour shading) according to their known
environmental preferences (see references in text): red: warm Atlantic water; orange: chilled Atlantic water; light blue: sea ice.






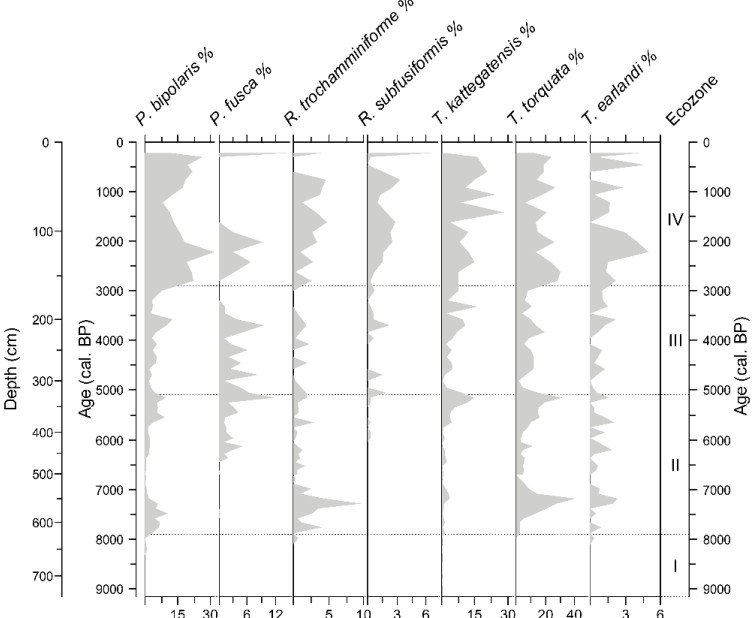


**Figure 5:** Down core distribution of the most abundant (>4% in at least one sample) agglutinated benthic foraminiferal species. Ecozones (I to IV) are given on the right side of the figure. Relative abundances are calculated based on the entire benthic (calcareous and agglutinated) foraminiferal assemblage.

### *Ecozone I: 9.2-7.9 cal. ka BP:*

This ecozone is highly dominated by the species *Stainforthia feylingi*, which contributes to almost 100 % of the benthic foraminiferal fauna. Only a few other species are represented here with abundances so low that they are considered insignificant. The foraminiferal concentrations are the lowest of the entire record, and planktonic specimens as well as agglutinated benthics are absent.

### *Ecozone II: 7.9-5.1 cal. ka BP*

The base of this ecozone is defined by a sudden increase in benthic species diversity and in both benthic and planktonic foraminiferal abundances. The abundance of *S. feylingi* decreases. Instead, *Cassidulina neoteretis, Cassidulina reniforme,* and *Islandiella norcrossi* show high abundances centred around 7.4 ka BP and again at 6 ka BP, separated by a very low abundance at 6.7 ka BP, coinciding with a general low species diversity and a temporary increase in *S. feylingi* and the common occurrence of *Bolivinellina pseudopunctata*. These two latter species combined constitute 70 % of the fauna at 6.7 ka BP. Overall the abundances of the two species groups made of *S. feylingi – B. pseudopunctata*, on one hand, and *C. neoteretis - C. reniforme – I. norcrossi*, on the other hand seem to be anti-correlated. Also noticeably is the significant abundance of up to 50



% of *Epistominella arctica* in the beginning of the ecozone. Characteristic for the end of the
ecozone is the large relative abundance of the agglutinated species compared to the calcareous
benthic fauna, again coinciding with a peak abundance of *B. pseudopunctata* and a drop in
frequencies of *C. neoteretis, C. reniforme,* and *I. norcrossi.* The most abundant agglutinated
species are *Portatrochammina bipolaris, Recurvoides trochamminiforme* and *Textularia torquata.*

### Ecozone III: 5.1-2.9 cal. ka BP

Overall, this ecozone is characterized by fluctuating abundances of many species. Both *Elphidium*
*clavatum* and *Nonionellina labradorica* show higher but fluctuating abundances compared to the
previous ecozone. The frequency of *E. arctica* peaks three times in this ecozone, reaching
abundances of around 30 %. Both *B. pseudopunctata* and *S. feylingi* display low abundances of
<1-5 % and 3-20 % respectively, while the decrease of *B. pseudopunctata* is very sudden in the
beginning of the ecozone. *C. neoteretis, C. reniforme* and *I. norcrossi* show a combined abundance
of 8-23 %. The relative frequencies of *Astrononion gallowayi* and *Buliminella elegantissima* tend
to be anti-correlated, with peak abundances of *A. gallowayi* in the beginning (7 %) and end (5 %)
of the ecozone corresponding to low (0 and 1 %, respectively) contributions of *B. elegantissima.*
The highest abundances of planktonic foraminifera for the entire core occurs in this ecozone at 3.2
ka BP. The abundance of agglutinated species is in general low but the frequency of
*Psammosphaera fusca* is relatively high, together with *Textularia kattegatensis* and *T. torquata.*

### Ecozone IV: 2.9-0.2 cal. ka BP

This ecozone is characterized by a sudden increase of the agglutinated/calcareous benthic species
ratio, as the agglutinated specimens outnumber the benthic calcareous individuals by a factor of
three. *P. bipolaris, T. kattegatensis* and *T. torquata* are among the most abundant agglutinated
species in this ecozone.  The dominance of agglutinated species coincides with a drop in the
contributions of planktonic foraminifera as well as of the benthic species *C. neoteretis, C.*
*reniforme* and *I. norcrossi.* The high abundances of agglutinated species persist towards the top of
the core, only interrupted by three periods of lower values at 1.6 ka BP, 1.2 ka BP and 0.8 ka BP,
corresponding to intervals with high contribution of *C. neoteretis* (6-8 %) and *C. reniforme* (6-8
%). *I. norcrossi* is in general poorly represented in this ecozone (< 1 %), while the percentage
frequency of *C. reniforme* is generally stable but lower than in ecozone II and III. *Epistominella*
*vitrea* experiences its highest mean relative abundance of the entire core within ecozone IV,
peaking at 1.2 ka BP (13 %). *E. clavatum, E. arctica* and *N. labradorica* abundances decrease





compared to the preceding ecozone and both *S. feylingi* and *B. pseudopunctata* are poorly
represented in this ecozone

**3.4 Geochemistry**
The XRF record shows several smaller events in addition to a general down-core pattern (Fig. 6).
Giraudeaux et al. (submitted) interpreted the elemental composition of this core in relation to
provenance of source sediments. Here we primarily focus on the terrestrial vs. marine signal.
Ecozone I is characterized by relatively low values of Br and Ca/Ti while the K and Rb counts are
high. Br counts increase throughout Ecozone II-III and become more or less stable in Ecozone IV.
The opposite pattern characterize the K and Rb counts. Both the Ca/Sr and Ca/Ti ratios are
relatively stable throughout the core; though, a slight increasing trend is seen in the Ca/Ti ratio
towards Ecozone IV. Both ratios show a prominent peak at around 6.7 ka BP in Ecozone II
coinciding with the highest values of IRD concentrations in the core.
We consider the element Br as an indicator of marine biological productivity often associated with
high amounts of marine organic matter (Pruysers et al., 1991). High counts of this element
therefore indicate minimal contribution of terrestrial-sourced material to the bulk sediment
(Calvert and Pedersen, 1993; Rothwell and Croudace, 2015). K and Rb are both typical for
environments with terrestrial influence (Saito, 1998; Steenfelt, 2001; Steenfelt at al., 1998). The
Ca/Ti and Ca/Sr ratios can be used as indicators of the marine biogenic origin of Ca (Bahr et al.,
2005; Richter et al., 2005). IRD counts and the mean grain size record are both indicators of
terrestrial influence, since larger grain sizes can be related to iceberg calving and or increased
sediment delivery by the Upernavik Isstrøm. More information about these two records is available
in Caron et al. (2018) and Giraudeau et al (submitted).

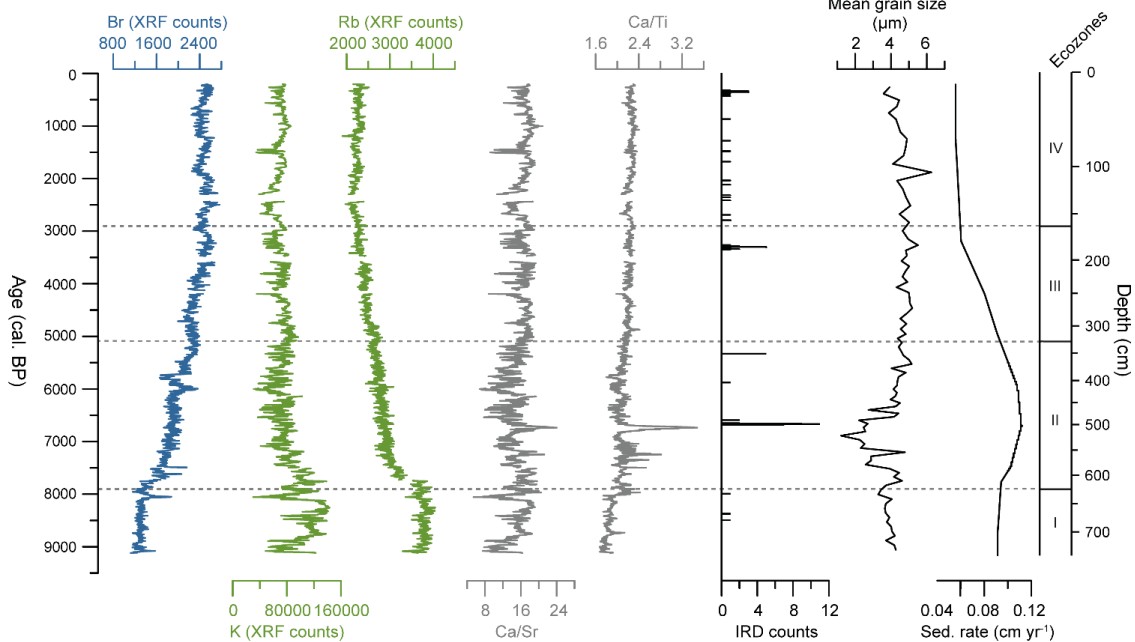

**Figure 6: From left to right:** X-ray fluorescence data, IRD concentration, mean grain size expressed in µm (Caron et al., 2018), and sedimentation rate in core AMD14-204C. The benthic foraminiferal ecozones are given in the right-most part of the plot. Gaps in data indicate missing data.

## 4 Paleoenvironmental interpretation

The distributional patterns of foraminiferal assemblages are indicators of changes in bottom and subsurface water conditions. Changes in the abundance ratio of agglutinated vs. calcareous specimens in sediments of Baffin Bay are often interpreted as evidence of subsurface deterioration, occasionally linked to the influx of the cold, saline and corrosive $CO_2$-rich Baffin Bay Deep Water (BBDW) (Jennings, 1993; Jennings & Helgadottir, 1994; Knudsen et al., 2008; Schröder-Adams & Van Rooyen, 2011). Off West Greenland, *I. norcrossi* and *C. neoteretis* are generally considered indicators of increased advection of Atlantic IC water into the WGC, based on their preference of relatively warm and high salinity waters (Knudsen et al., 2008; Perner et al., 2013; Seidenkrantz, 1995; Lloyd, 2006), albeit with *I. norcrossi* likely tolerating colder conditions and increased mixing with Polar water compared to *C. neoteretis*. *C. reniforme* has also been used as an indicator species for chilled Atlantic water, since it can live in somewhat colder and more saline water masses than the other Atlantic water indicator species presented here (Ślubowska-Woldengen et al., 2007). High abundances of *S. feylingi* and *B. pseudopunctata* are often considered associated with high primary productivity in the proximity of sea-ice edges; both species are tolerant to reduced bottom-water oxygen content (Knudsen et al., 2008; Seidenkrantz, 2013; Sheldon et al.,





2016). According to Seidenkrantz (2013), *S. feylingi* can be regarded as a typical sea-ice edge
indicator species. These micropaleontological proxy data, together with geochemical (XRF core
scanner-derived) and sedimentological data allow us to infer paleoenvironmental conditions within
each periods defined by the four foraminiferal ecozones.

*Ecozone I: 9.2-7.9 cal. ka BP:*
The total dominance of *S. feylingi* prior to 7.9 ka BP implies that conditions were unfavourable for
other foraminiferal species. *S. feylingi* is an opportunistic species, which can tolerate unstable low
oxygen conditions at the sea floor related to a stratified water column (Knudsen & Seidenkrantz,
1994; Patterson et al., 2000). The relatively high counts of the terrestrially-derived elements K, Rb
and a low Ca/Ti ratio, together with relatively high sedimentation rates (0.092 cm/year) could
indicate increased meltwater influence from the Greenland Ice Sheet. Furthermore, the low Br
counts and low absolute abundance of foraminifera imply that the general marine productivity was
low (Calvert and Pedersen, 1993; Pruysers et al., 1991). The absence of Atlantic water indicator
species suggests a weakening of the Atlantic water entrainment into the WGC, possibly in
connection with a WGC flow path located further away from the shelf. From 9.2 to 7.9 ka BP, the
eastern Baffin Bay region was therefore characterized by continuous meltwater injections from the
Greenland Ice Sheet (GIS) and an extensive sea-ice cover, associated with the final phase of the
deglaciation.
*Ecozone II: 7.9-5.1 cal. ka BP:*
The overall increase in species diversity from 7.9 ka BP indicates a transition towards ameliorated
subsurface conditions with higher marine biogenic productivity. The general decrease in Rb, K
and mean grain size together with increasing Br values point to a smaller influence of terrestrially-
derived sediment, possibly related to reduced meltwater inputs from the retreating Greenland Ice
Sheet.
These improved subsurface conditions were plausibly facilitated by a stronger entrainment of
Atlantic water masses into the WGC, inferred from the high contribution to the foraminiferal
assemblages of Atlantic water indicator species together with an increase in *P. bipolaris* which has
previously been linked to the presence of Atlantic water in the nearby Disko Bugt (Wangner et al.,
2018). The Atlantic water incursion seems especially strong at around 7.4 ka BP, coinciding with
an increase in planktonic foraminifera, indicative of increasing air temperatures and warming of



the (sub)surface waters, and further supported by the low abundances of the benthic sea-ice indicator species. Particularly the low abundance of *S. feylingi* coinciding with high percentages of *E. arctica* point to a reduction of the sea-ice cover, but high productivity (Seidenkrantz, 2013; Wollenburg & Mackensen, 1998).

The advection of Atlantic waters decreased significantly at 6.7 ka BP, as indicated by the sudden decrease in abundances of Atlantic water indicator species and a decrease in planktonic foraminifera. An increase in benthic sea-ice indicator species and an overall low benthic foraminiferal species diversity implies that the area was subjected to colder air temperatures, associated with an expansion of the sea-ice cover and a worsening in the subsurface conditions. Additionally, the transition towards higher abundance of benthic sea-ice species coincides with a large abundance peak of the agglutinated cold-water species *T. torquata* (Perner et al., 2012; Wangner et al., 2018). The peak values in the Ca/Ti Ca/Sr ratio around 6.7 ka BP suggest that a high amount of carbonate was exported to the area, possibly deposited as ice-rafted debris (IRD) according to the synchronous high IRD counts (Fig. 6). Previous studies have described the presence of detrital carbonate in the Baffin Bay, related to deposition by icebergs and or sea ice (e.g. Andrews et al., 2011; Jackson et al., 2017). This short-lived cold period at 6.7 ka BP can be related to a temporarily weaker incursion of Atlantic water off western Greenland, enabling cold Polar waters to enter the Baffin Bay, either in the form of increased EGC entrainment into the WGC and as Polar water delivered from the CAA. The event may potentially designate a very late meltwater event affecting the ocean circulation, but further investigations are needed to test this hypothesis.

At ca 6.0 ka BP, the Atlantic water contribution to WGC again increased, while sea ice retreated, based on the high frequency of the Atlantic water indicator species and the low abundance of sea-ice indicator species. The prevailing conditions were similar to those around 7.4 ka BP, but the lower abundances of the true Atlantic water indicator species *C. neoteretis* (cf. Seidenkrantz, 1995), implies that subsurface conditions were not as warm as around 7.4 ka BP.

The high agglutinated/calcareous foraminiferal ratio coinciding with low abundance of the Atlantic water indicator species just prior to 5.1 ka BP implies a short period of cold and corrosive subsurface waters, unfavourable for most of the calcareous benthic species. However, these conditions were favourable for the opportunistic benthic species *B. pseudopunctata*, which has been linked to environments with low oxygen conditions (Gustafsson and Nordberg, 2001;



Patterson et al., 2000). This deterioration of the subsurface environment can possibly be ascribed
to a decreasing strength of the WGC together with a presumably reducing Atlantic water
entrainment and a stronger influence of the cold corrosive BBDW.
*Ecozone III: 5.1-2.9 cal. ka BP*
A general amelioration of the bottom water environment and decreasing sea-ice cover, promoted
by a stronger Atlantic water entrainment at 5.1 ka BP, is suggested by an increased contribution
of Atlantic-water species and decreasing abundances of *B. pseudopunctata* and *S. feylingi*. High
contributions of *A. gallowayi* and *E. clavatum* imply that the hydrodynamic activity at the sea floor
was high and unstable in the beginning and end of the ecozone (Knudsen et al., 1996; Korsun &
Hald, 2000; Polyak et al., 2002), hereby related to a strengthening of the WGC flow.
The low abundances of *B. elegantissima* are possibly caused by the high turbidity levels. High
salinities linked to the strong entrainment of Atlantic derived water masses can also be inferred for
this time period considering the tolerance of *A. gallowayi* for raised salinity conditions (Korsun &
Hald, 1998). This fits well with the synchronous higher contributions of *C. reniforme*, which
previously has been associated with the incursion of chilled saline Atlantic waters (Ślubowska-
Woldengen et al., 2007).
The primary productivity species *N. labradorica* is often associated with the presence of fresh
phytodetritus in relation to primary productivity blooms and oceanic fronts (Jennings et al., 2004;
Polyak et al., 2002; Rytter, 2005). At our study site, this species seems to thrive under generally
warm bottom water conditions. *E. arctica* and *E. vitrea*, which are also both productivity indicators
(Perner et al., 2013; Scott et al., 2008; Wollenburg & Kuhnt, 2000; Wollenburg & Mackensen,
1998), show somewhat more fluctuating distributions in this ecozone, which could be linked to
shifting nutrient supply and fluctuating turbidity at the bottom. The overall high abundances of the
benthic productivity indicators reveal improved bottom water conditions with high food
availability.
*Ecozone IV: 2.9-0.2 cal. ka BP*
The sudden drop in calcareous foraminiferal concentrations illustrated by the very sudden increase
in the agglutinated/calcareous benthic ratio suggests that the decrease in the abundance of
calcareous specimens is most likely not a result of poor post-mortem preservation of these species
within the core, but rather related to environmental changes in the bottom waters. This is also
supported by the fact that the calcareous specimens are well preserved after 2.9 ka BP. We suggest





that the unfavourable conditions for the calcareous benthic foraminifera are associated with an
increasing influx of BBDW, impeding test formation of the calcareous species, because of the cold
corrosive property of this deep water mass. The increased inflow of BBDW was presumably
promoted by an overall weaker WGC flow and a diminishing entrainment of Atlantic water into
the WGC, as inferred from the phased decrease in abundance of Atlantic water indicator species.
Additionally, the lower sedimentation rate (0.056 cm/year) throughout this ecozone could possibly
be yielded by a weaker WGC flow strength. However, the continued, albeit lower, presence of
Atlantic water species and well-preserved calcareous specimens indicates some continued, at least
intermittent, influx of Atlantic water.
The short-term events of increased abundances of Atlantic water indicator species and high
planktonic foraminiferal concentrations centred roughly at 1.6, 1.2 and again at 0.8 ka BP are
possibly linked to periods of strengthening of the Atlantic water entrainment into the WGC,
resulting in short-term amelioration of the bottom and surface water conditions. The re-
strengthening of the WGC flow is supported by coinciding peak abundances of *A. gallowayi*
(Polyak et al., 2002). The productivity indicator species *E. vitrea* seems to favour conditions with
a relatively strong WGC possibly associated with the introduction of certain nutrients to the area.
Although the overall colder bottom water conditions could be expected to induce increased sea-
ice cover, conditions do not seem to have been favourable for the sea-ice indicator species *S.*
*feylingi* and *B. pseudopunctata*. However, these species are particularly thin-shelled and thus
highly sensitive to corrosive bottom water conditions.

**5 Discussion**

The interpretations of the benthic foraminiferal assemblage fauna and XRF data from this study,
suggest that several oceanographic and climatic changes, occurred during the Holocene in the
eastern Baffin Bay, associated with the relative change of Atlantic water mass advection, influence
of ice sheets, inflowing water masses derived from the Arctic Ocean, and the extent of sea-ice
cover. The changes herein are summarized in Fig. 7, with the number of planktonic foraminifera
and the sea-ice indicator species representing the surface water conditions and the
agglutinated/calcareous ratio represents fluctuations in deteriorating bottom water conditions
related to the incursion of colder, corrosive BBDW. The grouping of the Atlantic water
foraminifera was done following the methods of (Lloyd et al., 2011; Perner et al., 2012, Perner et
al., 2011), where *C. neoteretis*, *C. reniforme* and *I. norcrossi* were grouped, to represent the





alternation of Atlantic water mass advection to the eastern Baffin Bay. The percentage distribution
of the Atlantic-water group is represented by two curves. One calculated based on the combined
benthic foraminiferal assemblage including both agglutinated and calcareous species and one
without the agglutinated species. This was done in order to evaluate whether increases in this group
are driven by lower abundances of the agglutinated species. Additionally, the species *B.*
*pseudopunctata* and *S. feylingi* were grouped based on their preference of phytoplankton blooms
related to sea-ice margins. The Atlantic-water group is also represented by two different curves.
In Fig. 7, the summary curves from this study, are compared with the estimated mean July air
temperature, derived from regional pollen data from lake cores, using the Modern Analogue
Technique (Gajewski, 2015).

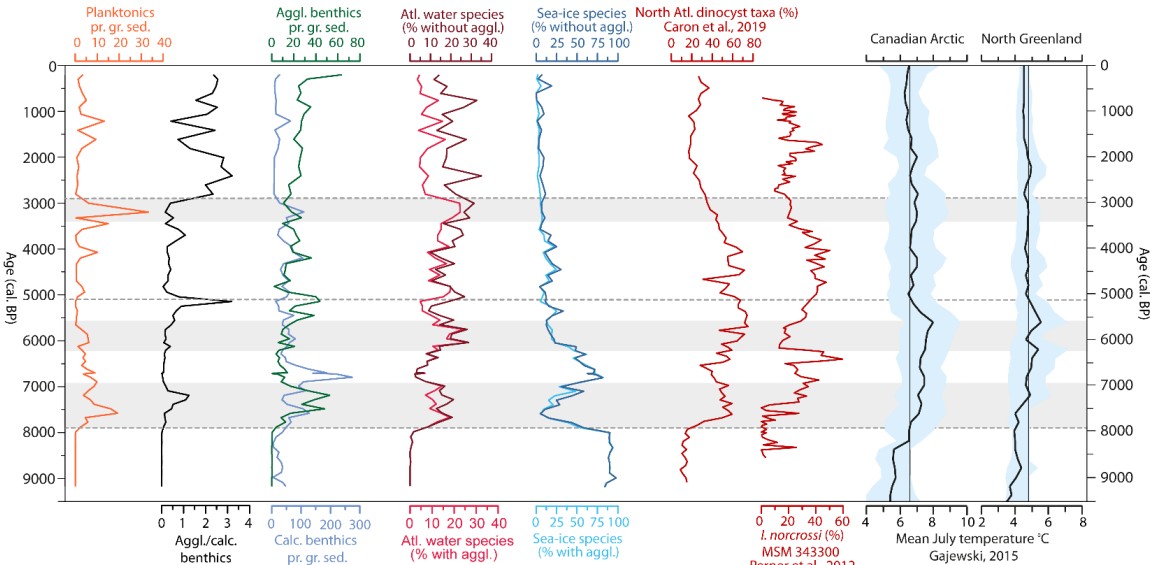


**Figure 7:** The green and purple curves show the comparison of the agglutinated benthics and calcareous benthics in individuals
per gram of wet sediment, respectively. The sea-ice indicator species curves represent a grouping of the two sea-ice indicator
species *S. feylingi* and *B. pseudopunctata*, shown in percentages including agglutinated species (light blue) and without agglutinated
species (dark blue). *C. neoteretis, C. reniforme* and *I. norcrossi* make up the Atlantic water indicator species shown in percentages
including agglutinated species (light red) and without agglutinated species (dark red). The grey bars represent periods of
strengthening of the WGC related to a stronger Atlantic water entrainment. The foraminifera data is compared to North Atlantic
dinocyst taxa (Caron et al., 2019) and the Atlantic water indicator species *I. norcrossi* from core MSM343300, Disko Bugt (Perner
et al., 2012). Additionally, two temperature reconstruction records are included, showing the mean regional July temperature (black
line) from selected sites, constructed by using the modern analogue technique (MAT) on pollen records from lake sediments
(Gajewski, 2015). The light blue shaded areas indicate the regional one standard deviations and the straight vertical line is the long-
term average of the curve.
**5.1 Early Holocene**
Several studies based on marine sediment cores from the Baffin Bay and adjacent areas, indicate
that this region was subjected to cold deglacial conditions during the earliest part of the Holocene.





A magnetic property study by Caron et al., 2018, carried out on core AMD14-204C, suggests that
the homogeneous clayey silts found from 9.2-7.7 ka BP and high values of $MDF_{NRM}$ and magnetic
susceptibility, represent a deglacial deposition dominated by glacially-derived material from an
ice-distal environment. These results are supported by studies of lake sediments adjacent to the ice
stream suggesting that the Upernavik Isstrøm had retreated close to its modern position (Briner et
al., 2013).
The strong influence of cold Polar waters from the Arctic Ocean and extensive sea ice that is
suggested by the dominance of *S. feylingi* and the low abundance of the North Atlantic dinocyst
taxa (Caron et al., 2019) (Fig. 7), is also observed further southwest of the core AMD14-204C site.
In the Labrador Sea (core MSM45-19-2 on Fig. 8), colder conditions were observed during the
period 8.9-8.7 ka BP, likely caused by increased advection of colder southward flowing Baffin
Bay water masses into the Labrador Current (Lochte et al., 2019). Additionally, the benthic
foraminiferal fauna indicate extreme conditions with low food supply and low oxygen conditions
related to an extensive sea-ice cover (Lochte et al., 2019) (Fig. 8). These environmental conditions
are further supported by dinocyst data from the eastern Baffin Bay west of Disko Bugt (core
CC70), indicating cold surface water conditions and extensive sea-ice cover prior to 9.5 ka BP
(Gibb et al., 2015). These data also show a shift towards slightly higher salinities and reduced sea
ice at ~9.5 ka BP, suggesting a decreasing influence of proximal ablation from the GIS (Gibb et
al., 2015).
The Disko Bugt in central West Greenland was subjected to similar cold conditions, where
sedimentological and benthic foraminiferal data from a marine sediment core near the Jakobshavn
Isbræ (core DA00-06) imply that the WGC influence was weaker and highly influenced by
significant meltwater influxes already prior to 8.3 ka BP (Lloyd et al., 2005). Additionally, a
second study from the Disko Bugt area (core MSM343300) document high abundances of Arctic
benthic foraminifera proposing a subsurface cooling, coinciding with increased meltwater
injection and sea-ice supply to the surface waters inferred from dinocyst, diatom and alkenone
($\%C_{37:4}$) data (Moros et al., 2016) (Fig. 8).
Accordingly, it seems that both surface and subsurface water conditions in the eastern Baffin Bay
and adjacent areas were highly affected by waning deglacial conditions in the Early Holocene,
with extensive sea-ice cover and ceasing meltwater influence from the marine outlet glaciers from
the GIS.



Reconstructed mean July temperatures based on pollen records from lake cores, point to colder
than average air temperatures during the Early Holocene in both the Eastern Canadian Arctic and
Northwest Greenland (a total of 13 sites) (Gajewski, 2015) (Fig. 7). This region was subjected to
cold air temperatures prior to 8.2 ka BP, due to the substantial remnants of the Laurentide Ice Sheet
(LIS), cooling the adjacent areas and supplying them with meltwater (Renssen et al., 2009). The
widespread stratification in West Greenland and in the Baffin Bay due to the increased meltwater
supply, is thought to have impeded the deep-water formation in the Labrador Sea (Renssen et al.,
2009; Seidenkrantz et al., 2013), resulting in a weaker northward flow of warmer air and water
masses (Renssen et al., 2009).

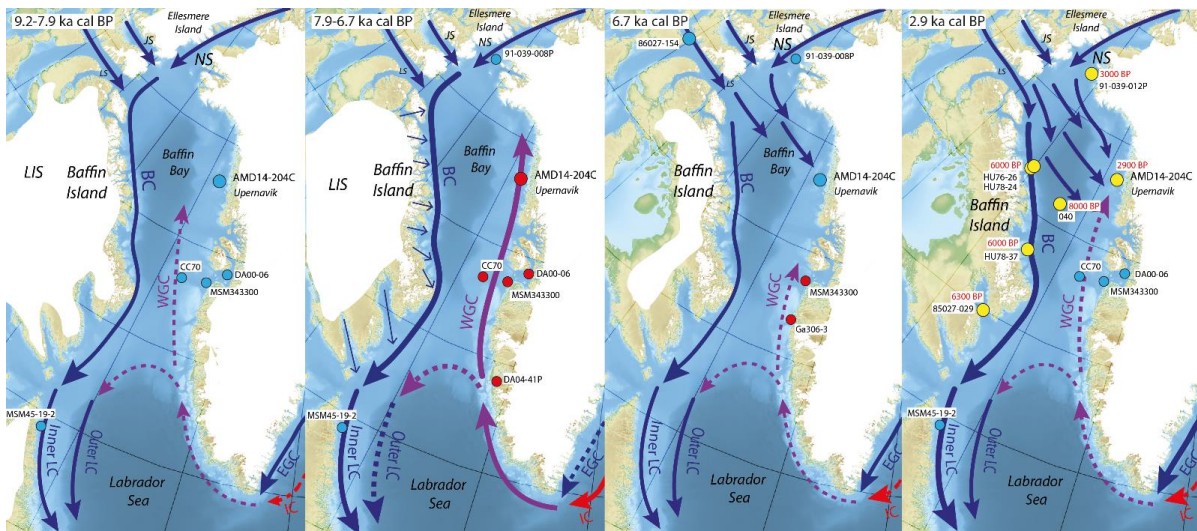


**Figure 8:** Map showing the oceanographic conditions in the Baffin Bay and Labrador Sea area from 9.2-2.9 ka BP based on this
core and other cores from the area; MSM45-19-2 (Lochte et al., 2019), CC70 (Gibb et al., 2015), DA00-06 (Lloyd et al., 2005),
DA04-41P (Seidenkrantz et al., 2013), MSM343300 (Perner et al., 2013, Moros et al., 2016), 91-039-008P (Levac et al., 2001),
86027-154 (Pieńkowski et al., 2014), 91-039-012P (Levac et al., 2001; Knudsen et al., 2008), HU76-26, HU78-24, HU-78-37
(Osterman & Nelson et al., 1989), 040 (Aksu, 1983), 85027-029 (Jennings, 1993). Red and blue cores represent relatively warmer
and colder conditions, respectively. Solid and dashed arrows indicate stronger and weaker ocean currents, respectively. The straight
blue arrows at Baffin Island at 7.9-6.7 ka BP indicate meltwater run-off into the ocean. The yellow cores at 2.9 ka BP indicate
sediment cores where a change towards an agglutinated dominated benthic fauna occurred where the red numbers indicate the
timing of this transition. Reconstruction of ice sheet extends are modified after Dyke et al., 2004. Abbreviations: LIS = Laurentide
Ice Sheet, LS = Lancaster Sound, JS = Jones Sound, NS = Nares Strait, BC = Baffin Current, LC = Labrador Current, IC = Irminger
Current, EGC = East Greenland Current, WGC = West Greenland Current.
**5.2 Mid Holocene**
The transition to warmer subsurface conditions was initiated around 7.9 ka BP at our study site,
marked by the increased abundances of Atlantic water indicator species in the benthic
foraminiferal assemblage, coinciding with low abundances of the sea-ice indicator species (Fig.





7). Additionally, the appearance of planktonic foraminifera and increase in the North Atlantic dinocyst taxa point to a warming of the surface waters (Caron et al., 2019). The warming of the subsurface waters in the eastern Baffin Bay seem to have persisted for most of the Mid Holocene (7.9-2.9 ka BP); however fluctuations in these conditions are evident. Benthic foraminiferal assemblage composition, and in particular the presence of *C. neoteretis*, infers that this temperature increase was caused by a strengthening of the WGC related to stronger entrainment of Atlantic water masses from 7.9-6.7 ka BP.

A concurrent shift in the oceanographic setting has also been identified west of Disko Bugt (core CC70, Fig. 8), where dinocyst assemblages imply increasing SST and further reduction of seasonal sea-ice cover from a strengthened Atlantic water inflow (Gibb et al., 2015). At the same site, the presence of benthic foraminiferal species associated with warm, subsurface water masses from 7.5 ka BP was likely also facilitated by decreased meltwater flow from the GIS together with increased inflow of Atlantic water mases (Jennings et al., 2014).

Southwest of Disko Bugt (core MSM343300; Fig. 8), evidence of warmer but variable subsurface water conditions is also here linked to  an enhancement of warm WGC influence, observed in the benthic foraminiferal record at 7.3-6.2 ka BP (Perner et al., 2012). At core site DA00-06 (Fig. 8) in Disko Bugt itself, a transition towards warmer conditions is marked by an increase in sub-arctic/Atlantic water benthic foraminifera after 7.8 ka cal. BP (Lloyd et al., 2005). This is further supported by the combined multiproxy study (core MSM343300; Fig. 8) by Moros et al., 2016, where low abundances of sea-ice diatoms and dinocysts indicate that also surface water conditions were warmer and relatively stable, with low meltwater influx from the Greenland ice sheet linked to warmer air masses in central West Greenland. A similar decreasing meltwater release from ca 7.5 ka BP is also seen further south in Ameralik Fjord near Nuuk (core DA04-41P; Fig. 8) (Seidenkrantz et al., 2013).

An oceanographic shift is also observed 7.3 ka BP in the Labrador Sea (core MSM45-19-2) that experienced decreasing surface and bottom water temperatures in connection to a strengthened northward flowing branch of the WGC compared to a weakened westward deflection of the WGC(Lochte et al., 2019; Sheldon et al., 2016). Surface-water reconstructions from the northernmost Baffin Bay (core 91-039-008P) and Newfoundland, i.e. path of the Baffin Current and Labrador Current, propose that increased advection of freshwater from melting Canadian Arctic glaciers strengthened the Baffin Current and Labrador Current (Levac et al, 2001; Solignac



et al., 2011). This shift in the flow of the warmer WGC causing an opposite pattern between the western Labrador Sea (core MSM45-19-2) and eastern Baffin Bay/central West Greenland (core CC70, MSM343300, DA00-06, DA04-14P; Fig. 8), was likely fostered by a strengthening of the subpolar gyre (SPG), as a result of the commencement of deep-water formation in the Labrador Sea at 7.5 ka BP (Hillaire-Marcel et al., 2001), after the strong meltwater fluxes from the GIS ceased. Warmer northward advection of Atlantic water masses along the coast of West Greenland, together with a stronger LC flow off eastern Canada, are both patterns typical for a strong SPG (Sheldon et al., 2016). The general Northern Hemisphere warming causing melting of Canadian Arctic glaciers and thus meltwater release to the Baffin Current and the Labrador Current would also strengthen this pattern (Solignac et al., 2011).

The generally warmer Mid-Holocene subsurface conditions at AMD14-204C were temporarily interrupted by a drop in the advection of warmer Atlantic water masses at 6.7 ka BP, where the abundances of the Atlantic water benthic foraminiferal indicator species decreased temporarily. Caron et al. (2018) observed a high IRD concentration at 6.7 ka (Fig. 6). It also coincides with low North Atlantic dinocyst taxa abundances (Caron et al., 2019), high sedimentations rates and a peak in the Ca/Ti and Ca/Sr elemental ratios together with high abundances of sea-ice indicator species in our study, suggesting overall cold surface and subsurface water conditions. Palaeozoic limestones and dolostones are commonly found at the flanks of Nares Strait and Lancaster Sounds in the northern part of Baffin Bay (Hiscott et al., 1989), whereas the northwestern coast of Greenland consists of fold belts consisting of reworked Archean basement rocks (mainly gneisses) interfolded with overlying sediment sequences (marble, schist and quartzite) and granitic intrusions (Henriksen, 2005). Older carbonate-rich layers are found in the Baffin Bay marine deposits as a result of ice-rafting in the northern Baffin Bay, which are then exported southward with the BC (Andrews et al., 2011). The IRD found in this core were presumably exported from the Nares Strait or Lancaster Sound by increased incursion of Polar water masses from the Arctic Ocean, transported southward by the BC, after which it re-circulated eastwards to the northeastern Baffin Bay, as has previously been suggested for older marine records (Andrews et al., 2011; Jackson et al., 2017). Adding to this, the eastward transport of IRD was possibly fostered by a strengthening of the northwesterly winds due to the decrease in high latitude insolation after 7 ka BP (Renssen et al., 2005). Supporting this, the Lancaster Sound was subjected to full cold Arctic conditions with enhanced sea-ice cover from 7.2-6.5 ka BP (Pieńkowski et al., 2014), and the Northern Baffin Bay experienced colder summer surface water temperatures (Fig. 8, core 91-039-





008P). However, in Disko Bugt there are no signs of surface and subsurface water cooling (Fig. 7)
(Moros et al., 2016; Perner et al., 2012; Erbs-Hansen et al., 2013), suggesting a local cooling of
the northern Baffin Bay.
A return to a period with warmer subsurface waters in the eastern Baffin Bay is facilitated by a re-
strengthening of the WGC and Atlantic water entrainment from 6.2-5.3 ka BP inferred by the
reappearance of high abundances in the Atlantic water indicator species in our study. The low
abundance of the Atlantic water indicator species *I. norcrossi* at around 6 ka BP in core
MSM343300 (Fig. 7) implies a cooling of the subsurface waters. However, the low abundance of
*I. norcrossi* here might have been caused by other factors such as changes in nutrients availability,
since other records in the Disko Bugt/central West Greenland area do not record a prominent
subsurface water cooling at that time (Erbs-Hansen et al., 2013; Jennings et al., 2014; Lloyd et al.,

624   2005).

Another drop in the WGC strength is evident at 5.3 ka BP at our study site, allowing the incursion
of both Polar surface waters and BBDW, as deduced by the high agglutinated/calcareous ratio
observed in this study. This event corresponds to the onset of a general decrease in the July air
temperatures over the Eastern Canadian Arctic (Fig. 7) (Gajewski, 2015), followed by generally
stable air temperatures above average until ca. 2.5 ka BP. The two periods with strong WGC flow
associated by enhanced Atlantic water incursion around 7.4 ka BP and again at 6.0 ka BP, seem to
occur simultaneously with increasing July air temperatures over the Eastern Canadian Arctic, (Fig.
7) (Gajewski, 2015).
The general subsurface conditions in the eastern Baffin Bay and West Greenland during the Mid
Holocene from 7.9 ka BP to ca. 2.9 ka BP are thus affected by overall warmer conditions, related
to a strong northward flow of Atlantic water masses, with minimal influx of meltwater from the
GIS. These warmer conditions coincide with the Holocene Thermal Maximum (HTM)
corresponding to the timing of the eastern Canadian Arctic (Kaufman et al., 2004), observed in
Greenland ice cores with peak warming at 7-6 ka BP (e.g. Dahl-Jensen et al., 1998; Johnsen et al.,
2001). The delayed onset of the HTM is in the eastern Canadian Arctic and eastern Baffin Bay
associated with the final collapse of the LIS (Kaufman et al., 2004).
**5.3 The Late Holocene**
The warm surface and subsurface conditions of the eastern Baffin Bay during the HTM, was
followed by a period of sudden deteriorating bottom-water conditions, as inferred from the abrupt



increase in the agglutinated/calcareous foraminiferal species ratio together with the presence of few Atlantic water indicator species and low abundances of planktonic foraminifera, attributed an enhanced BBDW advection to the core site. The green record in Fig. 7 shows that the distribution of agglutinated species does not increase significantly at the transition to this ecozone, whereas the abundance of the calcareous species (purple curve Fig. 7) drops abruptly. This implies that the increase in the agglutinated/calcareous ratio is not an artefact of a low abundance of agglutinated species down core due to bad preservation, but that it is in fact attributed a true oceanographic change. A marine sediment core from the southern Nares Strait, also recorded this abrupt shift towards a benthic foraminiferal fauna dominated by agglutinated species around 3.0 ka BP (Knudsen et al., 2008). The authors also explained this by an enhanced influence of Arctic Ocean water masses. Several studies from various parts of the Baffin Bay have in fact documented this increased Arctic Ocean water incursion but at various times with the earliest at 8 ka BP and the latest at ca. 3 ka BP (Aksu, 1983; Jennings, 1993; Osterman et al., 1985; Osterman & Nelson, 1989). Based on previous studies together with findings in our study, it can be deduced that the timing of the incursion of high saline, cold $CO_2$-rich Arctic water masses occurred in the deeper central part of the Baffin Bay first and later in the shallower coastal areas, as suggested by (Knudsen et al., 2008).

The cold BBDW does not reach the Disko Bugt at water depths greater than 300 m today (Andersen, 1981); however, cold conditions are also evident here. (Perner et al., 2012) recorded an increase in the abundances of agglutinated and Arctic water foraminifera at 3.5 ka BP, and they suggested that this was caused by a freshening of the bottom waters due to an increased entrainment of the EGC into the WGC, and a less significant Atlantic water entrainment. This agrees well with the low abundances of Atlantic water indicator species found in our study, possibly ascribed to a weaker AMOC. Concurrently, also the surface waters in Disko Bugt were cold in the Late Holocene (Moros et al., 2016), suggesting a general cooling trend of the subsurface and surface water temperatures in West Greenland (Andresen et al., 2011; Erbs-Hansen et al., 2013; Lloyd et al., 2007; Seidenkrantz et al., 2007; Seidenkrantz et al., 2008; Lloyd, 2006). An increased outflow of Polar waters from the Arctic Ocean, resulting in a strengthening and cooling of the Baffin Current and Labrador Current is documented in cores CC70 and MSM45-19-2 from the Labrador Shelf (Fig. 8), where dinocyst and benthic foraminiferal assemblages document a surface and subsurface water cooling after 3 ka BP (Gibb et al., 2015; Lochte et al., 2019). However, in the southwestern Labrador Sea, surface and subsurface water ameliorations are





recorded by dinocyst and benthic foraminifera data around 2.8 ka BP (Sheldon et al., 2016;
Solignac et al., 2011), indicating an increasing influence from warmer Atlantic water masses
versus the colder LC water masses, due to a northward placement of the frontal zone between the
Gulfstream and the LC (Sheldon et al., 2016), thus implying that the outflow of cold Arctic Ocean
waters did not reach the southeastern Labrador Sea.
The general cooling trend recorded in the marine records described here, is also observed in the
pollen records from the Eastern Canadian Arctic and North Greenland with July air temperatures
being lower than average starting at 1.5 and 2.8 ka BP, respectively (Fig. 7) (Gajewski, 2015).
This general cooling trend observed in vast areas of the North Atlantic in the late Holocene
corresponds to the Neoglaciation, linked to the initiation of readvances in many of the glaciers and
ice streams in West Greenland, including the Upernavik Isstrøm (Briner et al., 2013). An advance
of the Upernavik Isstrøm could explain the higher IRD counts in this ecozone, related to increased
iceberg calving.  However, it seems that the onset of the cold subsurface conditions in the eastern
Baffin Bay recorded in our study is not fully synchronous with the change towards colder summer
air temperatures in the Eastern Canadian Arctic. Nevertheless, the onset of the cold Neoglacial in
the eastern Baffin Bay resembles the onset of colder air temperatures recorded in North Greenland,
possibly related to the enhanced inflow of the cold Arctic water masses, subjecting the eastern
Baffin Bay to high latitude conditions alike the conditions in the North Greenland.
Superimposed on the Neoglacial cooling, shorter temporal subsurface water ameliorations are
evident in the eastern Baffin Bay, here associated to a re-strengthening in the WGC and Atlantic
water inflow, centred at 1.6 ka BP, 1.2 ka BP and 0.8 ka BP. These peaks in the Atlantic water
group are seen in both curves representing the percentage distribution of this group. However, the
percentages calculated without including the agglutinated species are quite high and not reliable
since the total sum of calcareous benthic foraminifera here are too low to be statistically significant
for interpretations.
In Disko Bugt the late Holocene is characterized by short-lived warmings of both the surface and
subsurface waters, related to an enhanced IC advection (Andresen et al., 2011; Lloyd, 2006; Moros
et al., 2006; Moros et al., 2016; Perner et al., 2012). Also records from the Labrador Sea have
documented these warmings from 2.0 to 1.5 ka indicated by fluctuating lengths of the sea-ice
seasons (Lochte et al., 2019), coinciding with shorter warmings found in the Placentia Bay in
Newfoundland (Solignac et al., 2011), and in the shelf waters of East Greenland (Jennings et al.,





2002). These widespread late Holocene centennial scale climate fluctuations were presumably facilitated by fluctuations in the atmospheric circulation pattern over the North Atlantic, controlling the strength of the northwesterly winds. However, a higher temporal resolution is needed in order to fully resolve these short-term climatic fluctuations documented in this study and other studies from the North Atlantic.

## 6 Conclusion

The presented multiproxy study based on benthic foraminiferal assemblage analysis and X-ray fluorescence data, document several climatic and oceanographic changes in eastern Baffin Bay during the Holocene:

1. The eastern Baffin Bay was subjected to cold deglacial conditions in the Early Holocene (9.2-7.9 ka BP) associated with an extensive sea-ice cover and meltwater inflows supplied by the melting of the Greenland Ice Sheet. Subsurface water conditions are characterized by a very low benthic foraminiferal species diversity and the coeval low abundances of Atlantic water indicator species reflecting a low entrainment of Atlantic water into the West Greenland Current.

2. A transition towards warmer subsurface water conditions is evident at the onset of the Mid Holocene (7.9 ka BP) encompassing the Holocene Thermal Maximum, where the eastern Baffin Bay was subjected to a strengthening in the West Greenland Current flow related to an increased Atlantic water incursion and ceasing meltwater influxes from the Greenland Ice Sheet. The ameliorating conditions found here are linked to a widespread oceanographic shift in the North Atlantic, due to the commencement of deep-water formation in the Labrador Sea.

3. The general ameliorating conditions found in the mid Holocene were interrupted by a cooling period centred at 6.7 ka BP, deduced from high abundances in the sea-ice indicator species and high IRD counts, where the latter presumably originated from the gateways of the Canadian Arctic Archipelago inferred by the high Ca-content observed in the XRF data. This cold period is ascribed to a weakening of the subpolar gyre, facilitating a weakening of the northward flowing Atlantic water masses along the West Greenland coast.

4. Evidence of enhanced inflow of the cold, corrosive and dense Baffin Bay Deep Water is documented at 5.3 ka BP, reflected by low abundances of the calcareous benthic species together with a decrease in the abundances of the Atlantic water indicator species. This is concurrent with a drop in the estimated July air temperatures found in the Eastern Arctic.





5. A drastic shift in the ocean circulation system occurred around 2.9 ka BP, ascribed to the onset of the Neoglacial cooling. The eastern Baffin Bay were subjected to an enhanced southward inflow of cold, corrosive and dense Baffin Bay Deep Water, recorded by the domination of agglutinated benthic foraminifera.

6. Short-lived bottom water warmings superimposed on the Neoglacial cooling, characterize the latest part of the Holocene, possibly facilitated by fluctuations in the atmospheric circulation system affecting the strength of the northwesterly winds.

## 7 Author contribution

M-SS developed the research idea. KEH conducted the benthic foraminiferal assemblage analysis with major contributions from M-SS. LW carried out the seven additional radiocarbon. JG provided four radiocarbon datings. CP performed the age modelling of the core. KEH prepared the manuscript with contributions from all co-authors.

## 8 Competing interests

Author M-SS is co-editor-in-chief of the journal.

## 9 Acknowledgment

We are grateful to the captain, crew and scientific party of the CCGS *Amundsen* 2014 expedition for their work in retrieval of sediment core AMD14-204C. Additionally, we would like to thank the ERC STG ICEPROXY 203441 for the financial support for the ship time. We also thank Guillaume Massé for the opportunity to work on the marine sediment core AMD14-204C. We also wish to thank Eleanor Georgiadis, Philippe Martinez, and Isabelle Billy for running the x-ray fluorescence spectroscopy of the core at the EPOC laboratory in Bordeaux, and for composing these datasets. We also thank Myriam Caron for the dynocyst, IRD and grain size data set. This study is a part of the "G-Ice" project, funded by the Danish Council for Independent Research (grant no. 7014-00113B/FNU) to MSS.

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
