# Peer review of "Reconstruction of Holocene oceanographic conditions in the Eastern"

_Climate of the Past, 2019_

## Referee Comment (RC1) · John Andrews (Referee) · 6 Feb 2020

This is a solid contribution to our increasingly detailed knowledge of changes in the oceanographic and glacial conditions in Baffin Bay and especially along the W Greenland margin. I do wonder about the continued use of a 140 ± 30 yr ocean reservoir correction without some statement or recognition that this undoubtedly changed throughout the Holocene as shown by several papers using the Iceland tephra as key markers (e.g. Ericsson, Kristjansdottir)—because of that it seems to me sensible to use a OCR=0 and a larger error estimate. That being said I doubt that this would make any substantial changes in their chronology or conclusions. Key aspects of the paper

are the plots (Figs. 4 and 5) of the downcore changes in the % of foraminifera (section 3.3). The paper identifies (p. 10) "...four ecozones...." Which are identified on thrse two figures. However, I saw no discussion on how these 4 zones were identified. Were they identified by the "eyeball" method, constrained clustering, or.....? The paleoenvironmental interpretation (Section 4) is primarily based on the foraminfera and I wonder would it not be more efficient to directly combine section 3.3 and 4 as the discussions of the Ecozones are inferring aspects of climate. In terms of the changes in the Holocene marine climate this paper, and I admit others that I have been a co-author on, neglect to mention the work in the 1970's and 1980's on the importance of marine mollusk faunas that were sampled and dated, often as part of efforts to date glacial isostatic uplift of the West Greenland and East Baffin Island coastal areas. I am thinking in particular of the 1st appearance of Mytilus edulis—the blue mussel (see refs below and references therein). On the east coast of Baffin Island Mytilus edulis invaded coastal waters ∼8.2 ka 14C (∼8.7 cal ka) and was present until about 3 ka . Another set of data that is worth looking at are the dates on the influx of wood to SW and W Greenland coasts carried around Greenland in the East Greenland and then West Greenland currents. These data would add important details that could be included on the summary figure Figure 8. On this figure and in the text they might consider what the effect might have been of the series of meltwater and sediment discharges through Hudson Strait (e.g. Barber et al., Jennings et al., ).

I often think that the detailed XRF-based geochemistry available is a method looking for answers. It would be interesting to compare the XRD mineral compositional data with the XRF data to gain a more detailed understanding of both (this was proposed and a method outlined by Eberl in the program "Hand Lense" USGS).

Conclusions: I enjoyed reading this paper. It provides valuable data to the growing body of literature documenting the complex of glaciological and oceanographic changes that effected the NW and W Greenland shelves and by inference the "downstream" margins of Baffin Island.

Andrews, J.T., Miller, G.H., Nelson, A.R., Mode, W.N., Locke, W.W., III, 1981. Quaternary near-Shore Environments on Eastern Baffin island, N.W.T., in: Mahaney, W.C. (Ed.), Quaternary Paleoclimate. Geo Books, Norwich, pp. 13-44. Barber, D.C., Dyke, A., Hillaire-Marcel, C., Jennings, A.E., Andrews, J.T., Kerwin, M.W., Bilodeau, G., McNeely, R., Southon, J., Moorehead, M.D., Gagnon, J.-M., 1999. Forcing of the cold event of 8200 years ago by catastrophic drainage of Laurentide lakes. Nature 400, 344-348.

Dyke, A.S., Dale, J.E., McNeely, R.N., 1996. MARINE MOLLUSCS AS INDICATORS OF ENVIRONMENTAL CHANGE IN GLACIATED NORTH AMERICA AND GREENLAND DURING THE LAST 18 000 YEARS*. Geographie physique et Quaternaire 50, 125-184. Dyke, A.S., England, J., Reimnitz, E., Jette, H., 1997. Changes in Driftwood Delivery to the Canadian Arctic Archipelago: The Hypothesis of Postglacial Oscillations of the Transpolar Drift. Arctic 50, 1-16. Jennings, A.E., Andrews, J.T., Wilson, L., 2015. Detrital Carbonate Events on the Labrador Shelf, a 13 to 7 kyr Template for Freshwater Forcing From the Laurentide Ice Sheet Quaternary Science Reviews 107, 62-80.

---

## Referee Comment (RC2) · Katrine Elnegaard Hansen et al. · 10 Mar 2020

The manuscript provides important new information on holocene ocean circulation
changes in eastern Baffin Bay, in an area where until now relevant knowledge has
been very limited. Thus, this is a significant contribution, fitting well within the scope of
CP. A few minor linguistic corrections are recommended, e.g. singular it/he/she = verb
+s, plural they = without adding s to verb

The work is based on the analysis of the benthic foraminiferal fauna of a 7.5 m long
sediment core in combination with sedimentological information and multi-element data
obtained from XRF scanning. Both the laboratory methods and data quality are of high
standard, whereas the age model is based on > 10 AMS C14 data levels well covering

the entire core.

Both titel and Abstract clearly and concise refer to the contents of the manuscript. Only doubt here, is whether the core site should be described as being situated in 'Northeastern' Baffin Bay, or this 'Northeastern' should be better replaced by just 'Eastern' (Baffin Bay).

As for the 'Introduction' I may suggest generally some shortening and re-structuring. This includes to move the section line 55 - 68 to the Regional Setting section, where some (double) information thus could be removed. Moreover, at the end of the 'Introduction, line 149 - 154, would fit better early in 'Regional Setting', i.e. following in line 95. More generally, some older references (oceanography/hydrography) could be omitted and/or replaced by more recent, f.ex.: *) Bi et al. 2019, Baffin Bay sea inflow and outflow..., The Cryosphere 13, 1025-1042; *) Castro de la Guardia et al. 2015. Potential positive feedback between Greenland Ice Sheet melt and Baffin Bay heat content on the west Greenland shelf. Geophys. Res. Lett. 42, 12; *) Munchow et al. 2015 Baffin Island and west Greenland Current Systems in northern Baffin Bay, Progr. in Oceanography.

Discussion: Within the context of the Ca/Ti and Ca/Sr interpretation, mid Holocene (line 595 - 608) another Ca source could be Uumannaq fjord area, where the Marmorilik Formation includes thick strata of dolomite marble and calcite marble ( Garde, 1979, Precambrian Research 8, 3-4, p.183-199). This possible source is found not far to the south, i.e. drifting with the WGC, icebergs from Uumannaq Fjord may (also) have contributed.

With regard to the Mid- and Late Holocene, a short reference should be made to the later HTM stage, where f.ex. in Ameralik, near Nuuk, evidence was found for (still) strong melting until 3.2 ka ( see Møller et al. 2006 Late Holocene environmental and climatic changes in Ameralik Fjord, Southwest Greenland – evidence form the sedimentary record. The Holocene 16, (5), 685-695). Same applies to Disko Bugt. Within

this context, important to note is significant cooling and freshening recorded (until c. 4.0 ka) in Newfoundland cores (e.g. Solignac, Sheldon), which must be related to Baffin Bay (melting, NAO-ocean control) conditions. Furthermore, the North Water Polynia was correctly mentioned in the Introduction; which function/contribution to corrosive bottom waters could this have had after it had formed ?

And finally: Great support for your conclusions you can find in Saini et al. 2020. Holocene variability in sea ice and primary productivity in the northeastern Baffin Bay, Arktos doi:10.1007/s41063-020-00075-y !

Herewith I may (thus) strongly support publication of this manuscript

—————————————————

---

## Author Comment (AC1) · 31 Mar 2020

**Author answers to reviewers comments on "Reconstruction of Holocene oceanographic conditions in the Northeastern Baffin Bay" by Katrine Elnegaard Hansen et al.**

**John Andrews (Referee)** andrewsj@colorado.edu

*We are very grateful to John Andrews for his effort and time to review our manuscript and for his highly beneficial comments and suggestions.*

This is a solid contribution to our increasingly detailed knowledge of changes in the oceanographic and glacial conditions in Baffin Bay and especially along the W Greenland margin.

I do wonder about the continued use of a 140 ± 30 yr ocean reservoir correction without some statement or recognition that this undoubtedly changed throughout the Holocene as shown by several papers using the Iceland tephra as key markers (e.g. Ericsson, Kristjansdottir)—because of that it seems to me sensible to use a OCR=0 and a larger error estimate. That being said I doubt that this would make any substantial changes in their chronology or conclusions.

*The reservoir age and its possible variability in the Holocene is indeed an important issue. We decided to use the same reservoir age correction as other similar studies in the area (e.g. Lloyd et al., 2011, Perner et al., 2012, Jackson et al., 2017), as described in the manuscript. Because of the potentially changing reservoir age in the Baffin Bay throughout the Holocene, we do not draw any vast conclusions on paleoceanographic and environmental changes down to a centennial time scale. We have added a sentence to the chronology methods description to acknowledge the possibility of a varying local reservoir age, see lines 174-176.*

Key aspects of the paper are the plots (Figs. 4 and 5) of the down core changes in the % of foraminifera (section 3.3). The paper identifies (p. 10) ". . .four ecozones. . .." Which are identified on those two figures. However, I saw no discussion on how these 4 zones were identified. Were they identified by the "eyeball" method, constrained clustering, or. . ...?

*We agree that this was not stated clearly in the manuscript. The four ecozones were indeed identified by visual interpretation of the species abundance and the boundaries were placed where major changes occurred in the benthic foraminiferal assemblage data. This has now been added to the revised manuscript, please see lines 236-239.*

The paleoenvironmental interpretation (Section 4) is primarily based on the foraminfera and I wonder would it not be more efficient to directly combine section 3.3 and 4 as the discussions of the Ecozones are inferring aspects of climate.

*This is definitely a possibility, but although the paleoenvironmental interpretation is indeed primarily based on changes in the benthic foraminiferal assemblages, it is also based on the XRF record of the core. Also, we prefer not to mix results and interpretation in the same section. Consequently, we have decided to keep sections 3.3 and 3.4 separated from section 4 to keep a clear distinction between the description of the results and the interpretation of the record.*

In terms of the changes in the Holocene marine climate this paper, and I admit others that I have been a co-author on, neglect to mention the work in the 1970's and 1980's on the importance of marine mollusk faunas that were sampled and dated, often as part of efforts to date glacial isostatic uplift of the West Greenland and East Baffin Island coastal areas. I am thinking in particular of the first appearance of Mytilus

edulis—the blue mussel (see refs below and references therein). On the east coast of Baffin Island Mytilus edulis invaded coastal waters ~8.2 ka 14C (~8.7 cal ka) and was present until about 3 ka.

Another set of data that is worth looking at are the dates on the influx of wood to SW and W Greenland coasts carried around Greenland in the East Greenland and then West Greenland currents. These data would add important details that could be included on the summary figure Figure 8. On this figure and in the text they might consider what the effect might have been of the series of meltwater and sediment discharges through Hudson Strait (e.g. Barber et al., Jennings et al.,).

*Thanks for pointing this out. The timing of the occurrence of M. edulis on east Baffin Island matches well with the interpretation of our foraminiferal assemblages. We have now included the reference to the marine mollusk studies to our discussion (see lines 553-559).*

*We have also added the role of the deflection of the Transpolar Drift on the pathways of Arctic Ocean waters i.e. via the gateways of the Canadian Arctic Archipelago (westward deflection) and via Fram Strait, derived from the distribution of driftwood. There is a correlation between more driftwood found in the CAA during the Neoglacial compared to Greenland, implying that the Transpolar Drift advected more cold Arctic Ocean water through the CAA, see lines 706-709.*

*We have added additional references from Jennings et al., (2015) and Barber et al., (1999) linked to meltwater introductions from the Laurentide Ice sheet to the Labrador Sea and thus affecting the deep-water formation here, see line 524-525.*

I often think that the detailed XRF-based geochemistry available is a method looking for answers. It would be interesting to compare the XRD mineral compositional data with the XRF data to gain a more detailed understanding of both (this was proposed and a method outlined by Eberl in the program "Hand Lense" USGS).

*Thank you for your suggestion. A XRD mineral compositional dataset could indeed be valuable, but we have not yet performed such analyses, and we believe it would not significantly alter the main conclusions of our paper. We will keep this in mind for a follow-up study.*

Conclusions: I enjoyed reading this paper. It provides valuable data to the growing body of literature documenting the complex of glaciological and oceanographic changes that effected the NW and W Greenland shelves and by inference the "downstream" margins of Baffin Island.

*Thank you!*

Andrews, J.T., Miller, G.H., Nelson, A.R., Mode, W.N., Locke, W.W., III, 1981. Quaternary near-Shore Environments on Eastern Baffin island, N.W.T., in: Mahaney, W.C. (Ed.), Quaternary Paleoclimate. Geo Books, Norwich, pp. 13-44.
Barber, D.C., Dyke, A., Hillaire-Marcel, C., Jennings, A.E., Andrews, J.T., Kerwin, M.W., Bilodeau, G., McNeely, R., Southon, J., Moorehead, M.D., Gagnon, J.-M., 1999. Forcing of the cold event of 8200 years ago by catastrophic drainage of Laurentide lakes. Nature 400, 344-348.
Dyke, A.S., Dale, J.E., McNeely, R.N., 1996. MARINE MOLLUSCS AS INDICATORS OF ENVIRONMENTAL CHANGE IN GLACIATED NORTH AMERICA AND GREENLAND DURING THE LAST 18 000 YEARS*. Geographie physique et Quaternaire 50, 125-184.
Dyke, A.S., England, J., Reimnitz, E., Jette, H., 1997. Changes in Driftwood Delivery to the Canadian Arctic Archipelago: The Hypothesis of Postglacial Oscillations of the Transpolar Drift. Arctic 50, 1-16.

Jennings, A.E., Andrews, J.T., Wilson, L., 2015. Detrital Carbonate Events on the Labrador Shelf, a 13 to 7 kyr Template for Freshwater Forcing From the Laurentide Ice Sheet Quaternary Science Reviews 107, 62-80. Interactive comment on Clim. Past Discuss., https://doi.org/10.5194/cp-2019-152, 2020.

---

## Author Comment (AC2) · 31 Mar 2020

**Author answers to reviewers comments on "Reconstruction of Holocene oceanographic conditions in the Northeastern Baffin Bay" by Katrine Elnegaard Hansen et al.**

**Anonymous Referee #2**

*We are very grateful to the anonymous referee #2 for his/her effort and time to review our manuscript and for his/her highly beneficial comments and suggestions.*

The manuscript provides important new information on Holocene ocean circulation changes in eastern Baffin Bay, in an area where until now relevant knowledge has been very limited. Thus, this is a significant contribution, fitting well within the scope of CP. A few minor linguistic corrections are recommended, e.g. singular it/he/she = verb +s, plural they = without adding s to verb.

*Thank you for making us aware of these mistakes. We have corrected the grammar where it was needed (lines 22, 77, 280, 375, 404, 502, 512, 548, 655).*

The work is based on the analysis of the benthic foraminiferal fauna of a 7.5 m long sediment core in combination with sedimentological information and multi-element data obtained from XRF scanning. Both the laboratory methods and data quality are of high standard, whereas the age model is based on > 10 AMS C14 data levels well covering the entire core.

Both title and Abstract clearly and concise refer to the contents of the manuscript. Only doubt here, is whether the core site should be described as being situated in 'Northeastern' Baffin Bay, or this 'Northeastern' should be better replaced by just 'Eastern' (Baffin Bay).

*We had chosen the term Northeastern because previous publications on the same record had done so (Caron et al 2018, 2019), but you are correct that location is not really north. We have changed northeastern to eastern where applicable, see lines: 1 (title), 59, 68, 112, 155, 613.*

As for the 'Introduction' I may suggest generally some shortening and re-structuring. This includes to move the section line 55 - 68 to the Regional Setting section, where some (double) information thus could be removed.

*Done.*

Moreover, at the end of the 'Introduction, line 149 - 154, would fit better early in 'Regional Setting', i.e. following in line 95.

*Thank you for your suggestions. We agree that the Upernavik Isstrøm should in fact be introduced earlier in the 'Regional settings' section, thus we have moved that section to the suggested place, see lines 80-85.*

More generally, some older references (oceanography/hydrography) could be omitted and/or replaced by more recent, f.ex.: *) Bi et al. 2019, Baffin Bay sea inflow and outflow..., The Cryosphere 13, 1025-1042; *) Castro de la Guardia et al. 2015. Potential positive feedback between Greenland Ice Sheet melt and Baffin Bay heat content on the west Greenland shelf. Geophys. Res. Lett. 42, 12; *) Munchow et al. 2015 Baffin Island and west Greenland Current Systems in northern Baffin Bay, Progr. in Oceanography.

*We have added the suggested references to the manuscript. See lines 78+100+111 (Münchow et al., 2015), 121 (Castro de la Guardia et al., 2015), 88+149 (Bi et al., 2019).*

Discussion: Within the context of the Ca/Ti and Ca/Sr interpretation, mid Holocene (line 595 - 608) another Ca source could be Uumannaq fjord area, where the Marmorilik Formation includes thick strata of dolomite marble and calcite marble ( Garde, 1979, Precambrian Research 8, 3-4, p.183-199). This possible source is found not far to the south, i.e. drifting with the WGC, icebergs from Uumannaq Fjord may (also) have contributed.

*Rink Isbræ from the Uumannaq fjord area may indeed have contributed with additional Ca-rich IRD as you mentioned. It has now been added to the discussion (Lines 619-621).*

With regard to the Mid- and Late Holocene, a short reference should be made to the later HTM stage, where f.ex. in Ameralik, near Nuuk, evidence was found for (still) strong melting until 3.2 ka ( see Møller et al. 2006 Late Holocene environmental and climatic changes in Ameralik Fjord, Southwest Greenland – evidence form the sedimentary record. The Holocene 16, (5), 685-695). Same applies to Disko Bugt. Within this context, important to note is significant cooling and freshening recorded (until c. 4.0 ka) in Newfoundland cores (e.g. Solignac, Sheldon), which must be related to Baffin Bay (melting, NAO-ocean control) conditions.

*Thank you for your suggestions on these papers. We have added additional information based on your suggested reference: Møller et al., 2006: Line 637-641. Though, the freshening and cooling in the Labrador Sea/Newfoundland (Solignac et al., 2011 and Sheldon et al., 2016) has already been discussed in the 'Mid Holocene' section (lines: 576-583 + 590-593).*

Furthermore, the North Water Polynya was correctly mentioned in the Introduction; which function/contribution to corrosive bottom waters could this have had after it had formed?

*As you point out, we discuss in the introduction that the North Water Polynya may play an important role in the formation of Baffin Bay Deep Water, but this is as mentioned not fully resolved for the present day. Thus, we have not been able to make any direct connection to NOW activity and periods of increased Baffin Bay Deep Water formation/advection to the study site.*

And finally: Great support for your conclusions you can find in Saini et al. 2020. Holocene variability in sea ice and primary productivity in the northeastern Baffin Bay, Arktos doi:10.1007/s41063-020-00075-y ! Herewith I may (thus) strongly support publication of this manuscript, https://doi.org/10.5194/cp-2019-152, 2020.

*Thank you for suggesting this paper. It has now been referenced in association with the peak in benthic sea-ice species recorded at 6.7 ka BP, where the biomarker record in Sanei et al., 2020 record a significant increase in the HBI III biomarker. See lines 601-604.*

**Notes:**

*Additional information was added based on the newly published paper 'Local and regional controls on Holocene sea ice dynamics and oceanography in Nares Strait, Northwest Greenland' by Georgiadis et al., (2020), Marine Geology, https://doi.org/10.1016/j.margeo.2020.106115. See lines 494-497 + 552-553. Core site AMD14-Kane2b described in this paper was added to Figure 8.*